# Promising Effects of Montelukast for Critically Ill Asthma Patients via a Reduction in Delirium

**DOI:** 10.3390/ph17010125

**Published:** 2024-01-18

**Authors:** Yuan Li, Meilin Zhang, Shengnan Zhang, Guoping Yang

**Affiliations:** 1Center of Clinical Pharmacology, The Third Xiangya Hospital, Central South University, Changsha 410013, China; 217212079@csu.edu.cn (Y.L.); zhangmeilin69@163.com (M.Z.);; 2Xiangya School of Pharmaceutical Sciences, Central South University, Changsha 410013, China

**Keywords:** montelukast, delirium, leukotriene, critical illness

## Abstract

**Background:** Montelukast (MTK), a potent antagonist of cysteinyl leukotriene receptor 1, has shown therapeutic promise for the treatment of neuropsychiatric disorders. Delirium, a common complication in critically ill patients, lacks effective treatment. This study aims to explore the impact of pre-intensive care unit (ICU) MTK use on in-hospital delirium incidence and, subsequent, prognosis in critically ill patients. **Methods:** A retrospective cohort study (n = 6344) was conducted using the MIMIC-IV database. After propensity score matching, logistic/Cox regression, E-value sensitivity analysis, and causal mediation analysis were performed to assess associations between pre-ICU MTK exposure and delirium and prognosis in critically ill patients. **Results:** Pre-ICU MTK use was significantly associated with reduced in-hospital delirium (OR: 0.705; 95% CI 0.497–0.999; *p* = 0.049) and 90-day mortality (OR: 0.554; 95% CI 0.366–0.840; *p* = 0.005). The association was more significant in patients without myocardial infarction (OR: 0.856; 95% CI 0.383–0.896; *p* = 0.014) and could be increased by extending the duration of use. Causal mediation analysis showed that the reduction in delirium partially mediated the association between MTK and 90-day mortality (ACME: −0.053; 95% CI −0.0142 to 0.0002; *p* = 0.020). **Conclusions:** In critically ill patients, MTK has shown promising therapeutic benefits by reducing the incidence of delirium and 90-day mortality. This study highlights the potential of MTK, beyond its traditional use in respiratory disease, and may contribute to the development of novel therapeutic strategies for delirium.

## 1. Introduction

Delirium is a common complication in critically ill patients, primarily characterized by attention disturbances, alterations in consciousness level, and changes in cognitive function [1]. Its occurrence rate varies from 18% to 70.4% among different types of ICU hospitalized patients [2,3,4]. Delirium often involves suffering for both patients and their families and is associated with increased mortality and poor long-term prognosis during hospitalization [5,6,7]. Regrettably, the exact pathological, physiological, and physiological mechanisms of delirium remain unclear, and effective targeted treatments or medications have not yet been identified [8]. Recent research has identified a number of triggers for delirium, including inflammation, oxidative stress, and exposure to centrally acting drugs.

Montelukast (MTK), a potent antagonist of cysteinyl leukotriene receptor 1 (CysLTR1), is widely used to treat asthma and allergic rhinitis. The systemic anti-inflammatory effects of MTK have long been of interest. Studies have found that MTK’s anti-inflammatory and antioxidant properties improve brain dysfunction and defects, and enhance the survival of neurons under oxidative conditions [9,10,11]. Research to date has shown that MTK may provide neuroprotection to alleviate central system disorders, such as Parkinson’s disease, Alzheimer’s disease, and epilepsy [12]. Therefore, pharmacologically, MTK has potential protective effects against delirium.

Worryingly, however, MTK has been reported to be associated with an increased risk of neuropsychiatric events (NAEs), which include symptoms such as insomnia, irritability, generalized anxiety, aggression, hallucinations, and hyperactivity, primarily observed in children [13,14]. Nonetheless, findings from the FDA’s sentinel study did not demonstrate an elevated risk of NAEs with MTK when compared to inhaled corticosteroids in patients aged 6 years and older, as well as in patients with a history of psychiatric disorders [15]. In addition, other well-designed retrospective studies have also failed to find this association [13,16,17]. These findings suggest that an increased risk of NAEs associated with MTK use is still a matter of debate and requires further investigation.

Therefore, we cannot solely rely on the available studies to determine the effect of MTK on the neuropsychiatric status of critically ill patients. The purpose of this study is to contribute to this ongoing discussion by exploring the potential role of MTK in critically ill patients. Specifically, we aim to investigate the association between pre-ICU MTK exposure and the incidence of delirium, as well as other outcomes, through a retrospective cohort study. By examining this association, we hope to gain valuable insights into the potential impact of MTK on the neuropsychiatric status and overall outcomes of critically ill patients. The knowledge gained from this study will not only contribute to our understanding of the effects of MTK, but will also provide important information in the search for therapeutic agents for delirium in critically ill patients.

## 2. Results

### 2.1. Baseline Information and Clinical Outcomes

A total of 6344 asthma patients admitted to ICU were included in this study, out of which 444 individuals were exposed to MTK prior to being admitted to ICU (as depicted in Figure 1). The baseline information can be found in Table 1. Overall, the asthmatic patients exposed to MTK prior to ICU admission were significantly older and heavier than the control group (exposure: median age 66.1 years, weight 85.3 kg vs. control: median age 63.5 years, weight 81.6 kg; *p* = 0.012, 0.015, respectively). The exposed group exhibited a greater proportion of females and white individuals (exposure: females 65.3%, white 69.8% vs. control: females 56.7%, white 64.1%; *p* < 0.001, *p* = 0.017, respectively). Furthermore, a greater proportion of patients in the exposed group were enrolled in Medicare compared with other insurance plans (exposure 50.0% vs. control 41.7%, *p* = 0.003). In addition, patients in the exposed group had higher rates of myocardial infarction and congestive heart failure, but the difference between the two groups was not significant. Moreover, patients in the exposed group had a lower severity of illness on the first day of ICU admission, as indicated by various illness scores, including CCI, SAPS II, APS III, and SOFA.

As shown in Table 1, there were no significant differences between the exposed group and the control group in regard to all the variables after PSM. The results revealed that pre-ICU exposure to MTK was significantly associated with a lower incidence of delirium during hospitalization (odds ratio (OR) after matching: 0.705; 95% CI 0.497–0.999; *p* = 0.049) and a lower 90-day mortality (OR: 0.554; 95% CI 0.366–0.840; *p* = 0.005). However, no significant association was observed between pre-ICU MTK exposure and mortality during hospitalization (OR: 1.087; 95% CI 0.607–1.946; *p* = 0.779). 

### 2.2. Subgroup Analyses

The effect of pre-ICU exposure on delirium demonstrated variation between the different subgroups (Figure 2A). The association was more significant in these patients: age ≤66 years (OR: 0.466; 95% CI 0.268–0.811; *p* = 0.006), females (OR: 0.636; 95% CI 0.411–0.985; *p* = 0.042), those not experiencing myocardial infarction (OR: 0.579; 95% CI 0.392–0.855; *p* = 0.006), those without diabetes (OR: 0.550; 95% CI 0.349–0.867; *p* = 0.009), CCI ≤ 3 (OR: 0.353; 95% CI 0.156–0.798; *p* = 0.010), and SAPS II scores of 24–39 (OR: 0.480; 95% CI 0.282–0.816; *p* = 0.006).

The effect of pre-ICU exposure on the 90-day mortality was more significant in patients who were female (OR: 0.560; 95% CI 0.344–0.912; *p* = 0.019), those not experiencing myocardial infarction (OR: 0.569; 95% CI 0.351–0.920; *p* = 0.020), and those with SAPS II scores of 24–39 (OR: 0.405; 95% CI 0.206–0.797; *p* = 0.007). Nevertheless, the notable aspect is that the association was more significant in patients aged > 66 years (OR: 0.476; 95% CI 0.286–0.794; *p* = 0.004). Furthermore, diabetes did not affect the association between pre-ICU exposure and the 90-day mortality.

### 2.3. Interaction Effects and Regression Models

Seven covariates, namely the type of ICU, SAPS II, SOFA, cerebrovascular disease, dementia, severe liver disease, and diabetes, which significantly affected the incidence of delirium were screened out by the forward stepwise selection method. We tested the significance of the interactions by introducing pre-ICU MTK exposure, possible interaction factors, and their interaction terms into the logistic regression model, along with the seven covariates. If the *p*-value for the interaction (shown in Figure 2) is less than 0.05, it indicated that the association between MTK exposure and delirium was significantly different across various strata of the variables examined. Ultimately, we found that myocardial infarction significantly attenuated the positive effect of MTK on delirium (the *p*-value for the interaction was 0.014).

Subsequently, a logistic regression model was developed to account for the impact of pre-ICU MTK exposure on delirium incidence, including the interaction term of pre-ICU MTK exposure and myocardial infarction. As shown in Table 2, for patients without myocardial infarction, the adjusted OR of the MTK exposure was 0.586 (95% CI: 0.383–0.896; *p* = 0.014), indicating that MTK had a significant positive effect on this subgroup. However, for patients with myocardial infarction, the adjusted OR of pre-ICU MTK exposure was 1.795 (95% CI 0.743–5.402), indicating that the association was not significant. The results from the regression model aligned with the results obtained from the subgroup analysis (shown in Figure 2A).

Interactions were also examined for the relationship between MTK exposure and 90-day mortality, and all *p*-values for the interaction were greater than 0.05 (Figure 2B). A Cox regression model was then developed to account for the relationship between pre-ICU MTK exposure and 90-day mortality, and the forward stepwise selection method was used to screen for covariates that significantly affected the 90-day mortality. The developed Cox regression model included the following variables: weight, CCI, APS III, LODS, cerebrovascular disease, paraplegia, severe liver disease, metastatic solid tumor, hemoglobin, hematocrit, platelet, blood urea nitrogen, and sodium. The adjusted HR of pre-ICU MTK exposure was 0.496 (CI 0.331–0.742, *p* < 0.001).

### 2.4. Further Analyses

As shown in Figure 3, the E-value for the point estimates regarding the association between pre-ICU MTK exposure and delirium incidence in patients without myocardial infarction was 1.94 (for 95% CI lower: 1.30). For the association between pre-ICU MTK exposure and 90-day mortality, the E-value for the point estimate was 3.45 (95% CI lower: 2.03). 

Causal mediation analysis demonstrated that the reduction in delirium by MTK partially mediated the reduction in 90-day mortality (ACME: −0.053; 95% CI −0.0142, −0.0002; *p* = 0.020; shown in Table 3).

As shown in Figure 2 and Appendix A, the long-term MTK subgroup showed a greater positive effect on delirium. However, the positive effect of MTK on 90-day mortality was not influenced by the duration of the pretreatment.

## 3. Discussion

The present study investigated for the first time the effect of MTK on the occurrence of delirium and in critically ill patients. In this retrospective cohort of critically ill patients with asthma, pre-ICU MTK exposure was found to be associated with reduced in-hospital delirium and 90-day mortality in both the original and matched cohorts. After subgroup analysis, we examined the interaction using regression models and found that the beneficial effect of MTK on delirium was more pronounced in patients without myocardial infarction and that the effect could be enhanced by the long-term use of MTK. Our causal mediation analysis demonstrated that the reduction in delirium by MTK partially mediated the reduction in the 90-day mortality. 

How might pre-ICU MTK exposure contribute to improved outcomes in critically ill patients? One possible hypothesis is that younger individuals, or those with fewer comorbidities and milder disease states, may show a preference for MTK, resulting in improved outcomes independent of MTK exposure. But, after PSM, the exposed and control groups were similar in terms of demographic characteristics, overall comorbidities, and disease severity at admission (as shown in Table 1), and the comparison of the outcomes still showed a significant association between MTK and reduced delirium incidence and 90-day mortality. Furthermore, the same conclusions were obtained after adjusting for comorbidity and disease severity using logistic and Cox regression models. Therefore, we can reject the hypothesis that the positive effect of MTK derives from the influence of bias and, instead, propose that MTK does reduce delirium incidence and mortality in critically ill patients through a number of possible pharmacological mechanisms. 

Delirium is a severe neuropsychiatric syndrome characterized by the acute onset of deficits in attention and other aspects of cognition; the pathogenesis of delirium is thought to involve neurobiological processes, such as neuroinflammation, cerebrovascular dysfunction, and impaired neural network connectivity [1]. Neuroinflammation is widely recognized as a causative factor of delirium [18]. Animal model studies have investigated the potential mechanisms by which inflammation disrupts brain function. Acute stressors, such as inflammatory trauma, surgery, infection, and sepsis, activate macrophages in tissues and monocytes in the blood, or interact directly with the cerebral vasculature to stimulate the secretion of IL-1, IL-1β, IL-6, the tumor necrosis factor (TNF), and prostaglandin E2 (PGE2) by brain endothelial and perivascular macrophages [19]. These inflammatory mediators are secreted across the blood–brain barrier or directly into the brain parenchyma via perivascular macrophages, inducing the brain to trigger microglia to produce pro-inflammatory cytokines, such as IL-1β, reactive oxygen species (ROS) [20], and reactive oxygen nitrogen species. This can directly affect astrocytes and neurons, leading to neuronal dysfunction and, even, neural network damage [21]. The stress overlay also causes damage to the vascular system, leading to endothelial and blood–brain barrier (BBB) damage [22].

Leukotrienes (LTs) are lipid mediators crucial in the pathophysiology of inflammation, which play a significant role in neuroinflammation. In the brain, cysteinyl leukotriene receptors (CysLTRs), such as CysLTR1 and GPR17, are expressed on microglial cells, neurons, stem cells, and precursor cells. LTs, through the activation of central LT receptors (CysLT1R, CysLT2R, GPR17), signal for pro-inflammatory responses, leading to the production of pro-inflammatory cytokines and reactive oxygen species (ROS). This cascade induces neuroinflammation, blood–brain barrier (BBB) disruption, enhances neurodegeneration, and inhibits neurogenesis [23]. Montelukast (MTK) can penetrate the BBB and, upon binding to central CysTL1R, inhibits leukotriene signaling, thereby reducing neuroinflammation and aiding in BBB reconstruction, exerting a protective effect on the brain. Additionally, in a murine middle cerebral artery occlusion brain injury model, researchers found that MTK improves surgery-induced brain damage by suppressing the expression and production of substances damaging to brain neurons, such as matrix metalloproteinases (MMPs), cytokines, and interleukin-6. It also prevents the disruption of brain endothelial junction proteins, thus protecting cerebrovascular function [24]. In summary, the mechanisms by which MTK protects brain neuronal function are diverse, and further research is needed to explore these mechanisms. MTK’s impact on central CysLTs receptors plays a positive role in reducing the incidence of delirium.

The role of CysLTs and their receptors in various central nervous system (CNS) disorders, including epilepsy and Alzheimer’s disease, is becoming increasingly apparent. For instance, Michael et al. observed that MTK improved memory, cognitive function, neuroinflammation, and apoptosis, with a more pronounced therapeutic effect in women [23]. Similarly, Zhao et al. reported that MTK reduced oxidative stress, cerebral infarct volume, brain atrophy, and neuronal loss, thereby improving behavioral function [25]. Thus, MTK may not only act as an anti-neuritis protection agent in the brain, but also maintain central nervous system function in critically ill patients by reducing oxidative stress levels and protecting the blood–brain barrier, thereby reducing the incidence of delirium and mortality. Further research is needed to investigate the intrinsic mechanisms of MTK pretreatment in reducing delirium during hospitalization in critically ill patients.

Given the reported association between MTK and an increased risk of NAEs, we were cautious about the effects of MTK on the neuropsychiatric status of critically ill patients and, therefore, did not formulate a hypothesis regarding the association between MTK and the incidence of delirium before conducting this study. Ultimately, the results from our study demonstrated that pre-ICU exposure was greatly associated with the reduced incidence of delirium, suggesting a positive effect on the neuropsychiatric status of critically ill patients. However, it is important to note that MTK-related NAEs have been reported more frequently in children to date, possibly related to the fact that MTK is more commonly used in children, and possibly because children’s brains are at an early stage of maturation and the therapeutic effects of MTK are accompanied by a strong modulation of specific processes in the central system, resulting in a greater likelihood of adverse drug reactions [26]. Unfortunately, the MIMIC-IV database used in this study did not include data on ICU patients under 18 years of age, and only patients with a median age of 66 years were included, limiting the generalizability of the findings to the pediatric population. Further data on children using MTK are needed to assess the safety of MTK in the pediatric population and to better understand its potential risks and benefits.

Furthermore, montelukast is frequently administered in conjunction with corticosteroids in clinical settings. Although our study did not investigate the impact of this combination on the incidence of delirium, the use of corticosteroids can heighten nerve centre excitability, and prolonged and excessive application can result in abnormal mental states and, in some cases, induce psychosis. There may be a relationship between corticosteroids and delirium. Another hypothesis for the pathophysiology of delirium is that persistent high cortisol levels due to acute stress induce and/or maintain delirium [18]. In a retrospective cohort study, the use of the corticosteroid dexamethasone was found to exacerbate delirium [27]. Based on current findings, it is not recommended to use MTK in combination with corticosteroids for patients with neurological disorders. However, further studies are needed to provide evidence. 

Previous studies conducted on COVID-19 patients have found that MTK diminished mortality by enhancing pulmonary function and oxygenation through its anti-inflammatory properties and the facilitation of bronchial expansion [28]. Consistent with these findings, our study also revealed a significant association between pre-ICU exposure and a reduction in mortality within 90 days of admission among critically ill patients with asthma. The difference is that our causal mediation analysis revealed that MTK reduced 90-day mortality partly via reduced delirium. However, it is important to acknowledge that this finding does not exclude the presence of other mediating factors. As shown in Table 3, the ACME of reduced delirium accounted for only 8.2% of the total effect, suggesting that other mediators are likely to be involved. These potential mediators may include MTK preventing respiratory failure through bronchodilation, thereby reducing the risk of death. Additionally, MTK might improve renal function and prevent renal failure, thereby exerting a positive impact on patient outcomes [29,30]. Further investigation is necessary to explore these hypotheses and identify other potential mediators.

As shown in Figure 2, myocardial infarction status had the most notable impact on the association between MTK and delirium in this study. Myocardial infarction has been identified as a risk factor for delirium and is associated with unfavorable patient outcomes [31,32,33]. The results from the subgroup analysis suggested that the benefits of MTK may not outweigh the associated risk in individuals with myocardial infarction. After controlling for confounding variables using the logistic regression model, the OR for MTK in patients with myocardial infarction was 1.795 (Table 3). However, it should be noted that the 95% confidence interval of OR for MTK encompassed 1 and the upper limit was large. Therefore, it is not plausible to infer a definite association between MTK and an elevated incidence of delirium in this subgroup, only that the effect of MTK was very insignificant in this population. Additionally, we observed a relatively attenuated significance of the association between MTK and reduced 90-day mortality in patients who were younger and had a less severe disease state on admission to the ICU. This phenomenon may be due to the inherently lower risk of mortality in these specific subgroups. Analysis of the subgroup of long-term MTK users showed that prolonged treatment with MTK increased its efficacy in regard to delirium, but did not affect its efficacy in reducing 90-day mortality. Further research is needed to investigate the dose–response relationship and to determine the optimal MTK dosing regimen that can maximize the beneficial effects of MTK in reducing the incidence of delirium.

In conclusion, our study reveals the remarkable association between pre-ICU MTK exposure and reduced delirium incidence and 90-day mortality in critically ill asthma patients, contributing to the growing evidence on the therapeutic benefits of MTK beyond its traditional use in respiratory disease. The mechanism of MTK as a CysLTR antagonist is a crucial consideration, suggesting the CysLTR may be a potential target for delirium treatment. Future research can delve into the impact of MTK on neuroinflammation, neuronal function, and cognitive impairment associated with delirium, providing insights into the underlying mechanisms. We hope that this study will draw attention to the potential of modulators of leukotriene pathways, such as MTK, in the treatment of delirium, and stimulate the development of novel therapeutic strategies to improve the outcomes of critically ill patients.

Our study has several limitations. Firstly, the retrospective, non-randomized and single-centre design restricts the generalizability of the findings in this study, and a definitive causal relationship between pre-ICU MTK exposure and the incidence of delirium and mortality needs to be established by further research with a prospective design. Additionally, residual and unmeasured confounders may still be present despite attempts to minimize them through propensity score matching and regression models. To assess the robustness of the observed association, E-values were used, which measure the minimum magnitude of unmeasured confounding required to nullify the association. However, unlike *p*-values, the determination of robustness cannot rely solely on surpassing a reference cut-off value. Therefore, the E-values are for reference only.

Secondly, the scarcity of ICD codes in the MIMIC-IV database poses a challenge to the reliability of the results. For example, the number of patients with allergic rhinitis identifiable by ICD codes in the MIMIC-IV database is so sparse that more than one hundred patients who used MTK prior to ICU admission could not be retrieved for the indication of medication. Therefore, asthma was included as a screening criterion to ensure comparability between the exposed and control groups and to facilitate propensity score matching. However, even when we use discharge notes combined with medication prescription records, in addition to ICD codes to help identify asthma, there is still no guarantee that the patients identified as having asthma are absolutely accurate and appropriate. Moreover, the use of ICD codes only allowed analysis of the occurrence of delirium during the hospital stay, without comprehensive information on the daily delirium status and duration of delirium, which may have affected the observed associations.

Furthermore, data integrity is a critical concern. Retrospective determination of pre-ICU MTK exposure relied on single-center prescription records, introducing the possibility of inaccuracies. We attempted to control for socioeconomic determinants of health by adjusting for insurance and race. Additional adjustment for area of residence, such as postcode, would further support the findings in this study, which are not secondary to differences in social determinants of health; however, the MIMIC-IV database does not contain this information. Although we adjusted for insurance and race to control for socioeconomic determinants of health, the lack of information on patients’ place of residence in the MIMIC-IV database hinders our ability to fully assess the impact of social determinants on the study results.

## 4. Materials and Methods

### 4.1. Data Source and Study Design

This retrospective observational cohort study was conducted utilizing data sourced from the Medical Information Mart for Intensive Care-IV (MIMIC-IV Version 2.2) database, which is affiliated with Beth Israel Deaconess Medical Center (Boston, MA, USA). The MIMIC-IV v2.2 database encompasses hospitalization information from ICU admissions at the Beth Israel Deaconess Medical Center between 2008 and 2019, involving 73,181 distinct ICU admissions by 50,920 patients, aged 18 years and above [34]. The Institutional Review Board of MIT and Beth Israel Deacon Medical Center granted approval to this database. Data extraction was carried out by the author ZSN, who successfully completed the web-based course exam of the National Institutes of Health (NIH) and received certification (certification number 52356447). Given the nature of this study, patient-informed consent was waived.

If a patient had more than one hospitalization, only the hospitalization corresponding to the first ICU record was taken into account for the analysis. Patients who had been prescribed MTK prior to ICU admission were assigned to the exposure group, and those who had not were assigned to the control group. Additionally, the exposure group excluded patients whose MTK prescriptions ended more than 365 days before their first ICU admission, as they may have ceased using MTK before being admitted to ICU. Meanwhile, the control group excluded patients who had received an MTK prescription within 90 days after ICU admission. The detailed flow chart is shown in Figure 1.

### 4.2. Data Extraction and Missing Value Processing 

Data extraction from the MIMIC IV database was performed using the Structured Query Language with PostgreSQL (version 15). The extracted dataset included several variables, including patient age on admission, sex, race, weight, type of ICU on admission, insurance type, in-hospital mortality, survival time, 90-day mortality, and laboratory parameters. The analysis also included certain disease scoring systems, such as the Charlson Comorbidity Index (CCI) [35], Simplified Acute Physiology Score II (SAPS II) [36], Acute Physiology and Chronic Health Evaluation III (APS III) [37], Sequential Organ Failure Assessment (SOFA) [38], Systemic Inflammatory Response Syndrome (SIRS) [39], and Logistic Organ Dysfunction System (LODS) [40].

Asthma status was ascertained by ICD codes. Moreover, patients who had the keyword “asthma” in their discharge records and prescriptions for asthma medications (Appendix A) during hospitalization were also identified as having asthma. Delirium during hospitalization was assessed using the Confusion Assessment Method by the ICU. We utilized this approach, searching all the ICD codes associated with included subjects for the following terms: (1) delirium, (2) encephalopathy, (3) altered mental status, (4) confusion, (5) mental disorder, (6) hallucination, (7) delusion, and (8) cognitive impairment/deficit [41,42,43,44]. 

The survival time and 90-day mortality rates were determined by leveraging an existing cross-reference between the ICU admission time in the MIMIC-IV database and the Social Security Administration Death Master File. The laboratory parameters included in the analysis were the initial measurements taken when the patient was admitted to the ICU. The percentage of missing values for all laboratory parameters is less than 5%. An appropriate approach was used to deal with these missing values: if the particular parameter followed a normal distribution (such as hemoglobin, sodium concentration, etc.), the missing value was approximated by the mean; if it did not follow a normal distribution (such as serum creatinine, blood glucose value, etc.), the missing value was approximated by the median. Patient weight data also had missing values and outliers. To address missing data in this area, we first stratified the patients by race and sex and, then, estimated and imputed missing and outlier values using the median within each subgroup.

### 4.3. Statistical Analysis

Statistical analyses were performed using SPSS (version 28.0) and R (version 4.2.1), and *p* < 0.05 was considered statistically significant. Propensity score matching (PSM) was used to adjust for confounders between the two cohorts, using one-to-one nearest neighbor matching, with a caliper width of 0.01 and a random seed of 123.

Subgroup analyses were performed based on age, sex, comorbidities, and severity of illness at the time of ICU admission in our study. Subsequently, the forward stepwise selection method was used to screen out covariates that significantly affected the outcomes. Interaction *p*-values were estimated using the regression model, and interactions were considered present if the adjusted *p*-value was <0.05, indicating that the association between MTK and the outcome varied significantly between the strata [45]. Finally, potential confounders and covariates, and potential interaction terms were included in the logistic/Cox regression model to determine the association between pre-ICU MTK exposure and delirium and 90-day mortality.

### 4.4. Further Analyses

E-value sensitivity analysis was conducted to assess the minimum magnitude of unmeasured confounding required to nullify the observed associations [46]. The E-value typically ranges from 1 to infinity, with a higher E-value indicating a greater extent of unmeasured confounders necessary to negate the association between pre-ICU MTK exposure and the outcomes [47].

Causal mediation analysis was conducted to investigate whether the prevalence of delirium acts as a mediator between pre-ICU MTK use and mortality. The average direct effect (ADE) and average causal mediation effect (ACME) were calculated using the mediation R package.

The effect of MTK may be influenced by the dose and duration of its use in patients. However, the vast majority of MTK prescription records (99.24%) in the MIMIC-IV database were for a standard dose of 10 mg/day. This indicates that the total amount of MTK administered to the majority of patients is primarily determined by the duration of treatment rather than variations in dosage. The long-term medication subgroup was selected on the basis of the initial exposure group. Inclusion criteria were a record of at least two prescriptions for MTK prior to ICU admission, with the earliest prescription ending more than 90 days before the patient’s first ICU admission. After screening for long-term MTK users, we performed the PSM as described above for the statistical analysis.

## 5. Conclusions

In this study, we discovered that pre-ICU exposure to MTK was associated with reductions in in-hospital delirium and 90-day mortality, and that the favorable effect of MTK on delirium was more pronounced in patients without myocardial infarction, an effect that was potentiated by prolonged MTK use. Finally, our causal mediation analyses showed that the reduction in delirium by MTK partially contributed to the reduction in 90-day mortality. The findings from this study suggest that MTK may have potential beyond its traditional use in the treatment of respiratory disease. The mechanism of MTK as a CysLTR antagonist is a crucial consideration, indicating that the CysLTR may be a potential target for delirium treatment. Future research could investigate the impact of MTK on neuroinflammation, neuronal function, and cognitive impairment associated with delirium, providing insights into the underlying mechanisms. This study aims to draw attention to the potential of modulators of leukotriene pathways, such as MTK, in the treatment of delirium.

## Figures and Tables

**Figure 1 pharmaceuticals-17-00125-f001:**
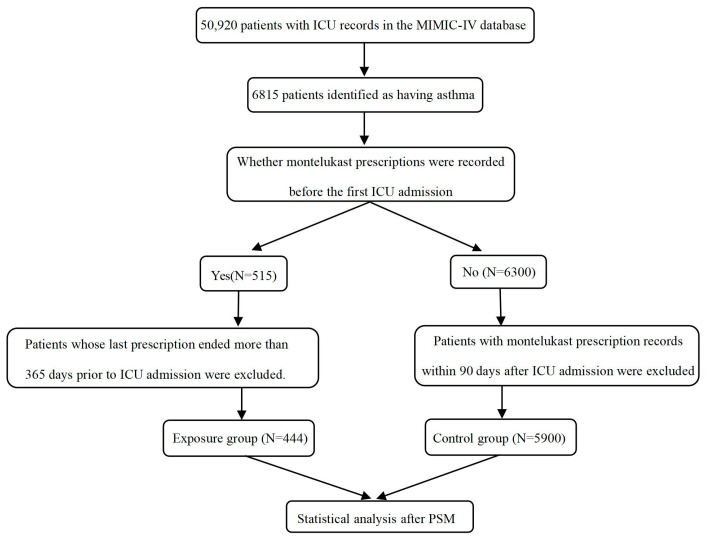
Flow diagram of patient selection process.

**Figure 2 pharmaceuticals-17-00125-f002:**
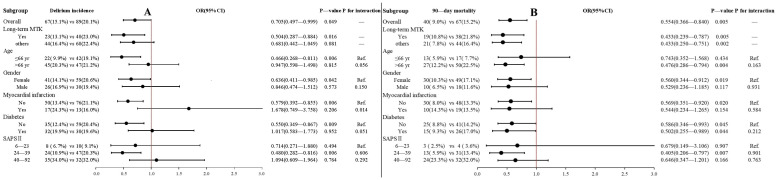
Forest plot for subgroup analyses. (**A**): The association between pre-ICU MTK exposure and incidence of delirium during hospitalization in different subgroups. The significance of the OR was evaluated in each subgroup using the chi-squared test, with a threshold of *p* < 0.05 considered statistically significant. The significance of the interactions was assessed by incorporating MTK exposure, potential interaction factors, and their interaction terms into the logistic regression model. In addition, seven covariates that significantly affected the occurrence of delirium selected by the stepwise forward method were added to the model, namely the type of ICU, SAPS II, SOFA, cerebrovascular disease, dementia, severe liver disease, and diabetes. (**B**): The association between pre-ICU MTK exposure and 90-day mortality in different subgroups. The significance of the interactions was assessed using the Cox regression model and the following covariates that significantly affected the 90-day mortality, selected by the stepwise forward method, were added to the model: weight, CCI, APS III, LODS, cerebrovascular disease, paraplegia, severe liver disease, metastatic solid tumor, hemoglobin, hematocrit, platelet, blood urea nitrogen, and sodium.

**Figure 3 pharmaceuticals-17-00125-f003:**
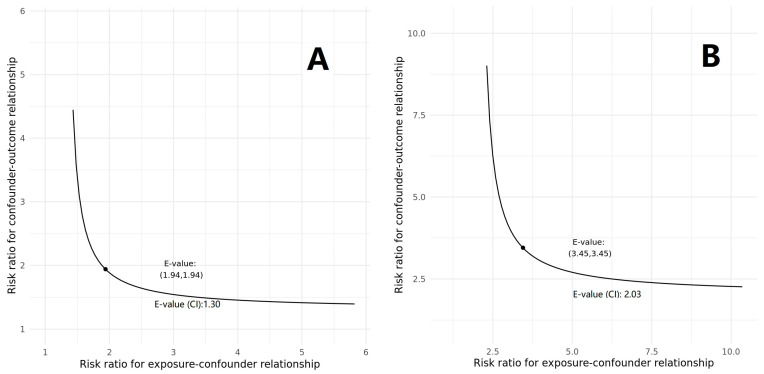
E-value for point estimation and the lower 95% CI. (**A**): Sensitivity of the association between pre-ICU MTK exposure and delirium during hospitalization in patients without myocardial infarction. (**B**): Sensitivity of the association between pre-ICU MTK exposure and 90-day mortality in critically ill patients with asthma.

**Table 1 pharmaceuticals-17-00125-t001:** Baseline characteristics and outcomes of patients in the original and matched cohorts.

Characteristics	Before Matching				After Matching	
	All Patients (n = 6344)	Exposure Group (n = 444)	Control Group (n = 5900)	*p* Value	Control Group (n = 443)	*p* Value
**Age (y)**	63.8 (51.8–75.0)	66.1 (54.7–74.6)	63.5 (51.4–75.0)	0.012	66.2 (54.5–77.4)	0.560
**Weight (kg)**	81.8 (68.4–98.0)	85.3 (70.0–102.5)	81.6 (68.2–98.0)	0.015	83.4 (68.9–103)	0.709
**Gender, n (%)**				<0.001		0.905
Female	3637 (57.3)	290 (65.3)	3347 (56.7)		287 (64.9)	
Male	2707 (42.7)	154 (34.7)	2553 (43.3)		155 (35.1)	
**Race, n (%)**				0.014		0.817
White	4089 (64.5)	310 (69.8)	3379 (64.1)		300 (67.9)	
Black	967 (15.2)	67 (15.1)	900 (15.3)		72 (16.3)	
Other ^a^	1288 (20.3)	67 (15.1)	1221 (20.7)		70 (15.8)	
**Type of insurance, n (%)**				0.003		0.188
Medicaid	617 (9.7)	36 (8.1)	581 (9.8)		39 (8.8)	
Medicare	2681 (42.3)	222 (50.0)	2459 (41.7)		194 (43.9)	
Other ^b^	3046 (48.0)	186 (41.9)	2860 (48.5)		209 (47.3)	
**Type of ICU for first admission, n (%)**				0.012		0.600
CCU/CVICU	2225 (35.1)	184 (41.4)	2041 (34.6)		178 (40.3)	
MICU/SICU	2337 (36.8)	143 (32.2)	2194 (37.2)		156 (35.3)	
Others ^c^	1782 (28.1)	117 (26.4)	1665 (28.2)		108 (24.4)	
**Comorbidity, n (%)**						
Myocardial infarction	887 (14.0)	70 (15.8)	817 (13.8)	0.261	81 (18.3)	0.311
Congestive heart failure	1733 (27.3)	132 (29.7)	1601 (27.1)	0.237	123 (27.8)	0.532
Peripheral vascular disease	629 (9.9)	44 (9.9)	585 (9.9)	0.997	48 (10.9)	0.643
Cerebrovascular disease	772 (12.2)	38 (8.6)	734 (12.4)	0.016	40 (9.0)	0.796
Dementia	154 (2.4)	5 (1.1)	149 (2.5)	0.065	5 (1.1)	0.994
Chronic pulmonary disease	4756 (75.0)	407 (91.7)	4349 (73.7)	<0.001	401 (90.7)	0.620
Rheumatic disease	260 (4.1)	32 (7.2)	228 (3.9)	<0.001	26 (5.9)	0.425
Peptic ulcer disease	139 (2.2)	8 (1.8)	131 (2.2)	0.561	10 (2.3)	0.627
Mild liver disease	711 (11.2)	27 (6.1)	684 (11.6)	<0.001	31 (7.0)	0.575
Diabetes	1994 (31.4)	161 (36.3)	1833 (31.1)	0.023	153 (34.6)	0.609
Paraplegia	220 (3.5)	12 (2.7)	208 (3.5)	0.361	13 (2.9)	0.830
Renal disease	1243 (19.6)	61 (13.7)	1182 (20.0)	0.001	56 (12.7)	0.638
Malignant cancer	741 (11.7)	62 (14.0)	679 (11.5)	0.120	46 (10.4)	0.106
Severe liver disease	260 (4.1)	9 (2.0)	251 (4.3)	0.022	9 (2.0)	0.992
Metastatic solid tumor	360 (5.7)	26 (5.9)	334 (5.7)	0.864	21 (4.8)	0.463
Aids	51 (0.8)	3 (0.7)	48 (0.8)	0.970	3 (0.7)	1.000
**Scores**						
CCI	5 (3–7)	5 (3–7)	5 (3–7)	0.127	5 (3–7)	0.722
SAPS II	31 (23–40)	31 (23–39)	32 (23–40)	0.059	31 (24–38)	0.443
APS III	38 (28–51)	35 (27–47)	38 (28–51)	0.003	36 (28–48)	0.407
SOFA	3 (2–6)	3 (1–5)	3 (2–6)	0.001	3 (2–5)	0.314
SIRS	3 (2–3)	3 (2–3)	3 (2–3)	0.005	3 (2–3)	0.494
LODS	3 (2–5)	3 (2–5)	3 (2–5)	0.009	3 (2–5)	0.824
**Laboratory tests**						
Hemoglobin concentration, g/dL	11.1 ± 2.2	10.9 ± 2.1	11.1 ± 2.2	0.131	10.9 ± 2.2	0.926
Hematocrit (%)	33.3 ± 6.0	33.0 ± 5.7	33.3 ± 6.0	0.209	32.9 ± 5.8	0.887
Platelet, K/μL	206.5 (152.5–271.0)	218.3 (157.6–285.1)	206.5 (152.0–271.0)	0.009	224 (160–287.8)	0.619
WBCs, K/μL	10.8 (7.9–14.5)	11.7 (8.5–14.9)	10.8 (7.9–14.5)	0.017	11.2 (8.2–15.3)	0.815
Bicarbonate, mEq/L	23.7 ± 4.3	24.6 ± 4.6	23.6 ± 4.3	<0.001	24.4 ± 4.3	0.510
BUN, mg/dL	17.5 (12.0–27.5)	17.0 (12.0–23.5)	17.5 (12.0–27.5)	0.038	17.0 (12.0–24.6)	0.835
Calcium, mg/dL	8.4 ± 0.8	8.5 ± 0.6	8.4 ± 0.8	0.445	8.5 ± 0.7	0.835
Creatinine, mg/dL	0.9 (0.7–1.3)	0.9 (0.7–1.2)	0.9 (0.7–1.4)	0.002	0.9 (0.7–1.2)	0.629
Glucose, mg/dL	128.0 (108.0–160.0)	128.5 (109.1–159.4)	128.0 (108.0–160.0)	0.914	128.0 (108.8–163.5)	0.878
Sodium, mEq/L	138.2 ± 4.4	137.9 ± 4.1	138.2 ± 4.4	0.131	138.0 ± 4.4	0.584
Potassium, mEq/L	4.2 ± 0.6	4.2 ± 0.6	4.2 ± 0.6	0.945	4.3 ± 0.6	0.601
**Outcomes, n (%)**						
Delirium	1235 (19.5)	67 (15.1)	1168 (19.8)	0.016	89 (20.1)	0.049
Hospital mortality	439 (6.9)	25 (5.6)	414 (7.0)	0.267	23 (5.2)	0.779
90-day mortality rate	820 (12.9)	40 (9.0)	780 (13.2)	0.011	67 (15.2)	0.005

Abbreviation: SICU, surgical intensive care unit; MICU, medical intensive care unit; CCU, coronary care unit; CVICU, cardiac vascular intensive care unit; WBCs, white blood cells; BUN, blood urea nitrogen; CCI, Charlson Comorbidity Index; SAPS II, Simplified Acute Physiology Score II; SAPS III, Simplified Acute Physiology Score III; SOFA, Sequential Organ Failure Assessment; SIRS, Systemic Inflammatory Response Syndrome; LODS, Logistic Organ Dysfunction System. Continuous variables were reported as mean ± standard deviation (SD), or median of the first quartile to the third quartile, depending on whether they followed a normal distribution. Categorical variables were presented as numbers and proportions. Comparisons were performed using the *t*-test, Mann–Whitney U-test, or chi-squared (or Fisher’s exact) test, according to the distribution characteristics of the data. ^a^ The category of “other races” included patients who identify as Hispanic/Latino, American Indian, Native Hawaiian, or other Pacific Islander, Portuguese, or Asian, as well as patients who opted not to disclose their race. ^b^ The category of “other type of insurance” included patients who did not utilize Medicaid or Medicare during their hospitalization. ^c^ The category of “other type of ICU” included trauma and neurology ICU.

**Table 2 pharmaceuticals-17-00125-t002:** Parameters of the logistics model incorporating the interaction term of pre-ICU MTK exposure and myocardial infarction.

Variable	Coefficient	Standard Error	OR (95% CI)	* *p*-Value
Pre-ICU MTK exposure	β1: −0.534	0.217	0.586(0.383–0.896) ^b^	0.014
Interaction item	β2: 1.119 ^a^	0.489	-	0.022 ^a^

*: With a *p*-value of less than 0.05, the results are statistically significant, indicating that the results are unlikely to be due to random factors. Other covariates: race, age, gender, type of ICU, SAPS II, SOFA, myocardial infarction, cerebrovascular disease, dementia, severe liver disease, and diabetes. ^a^ The regression coefficient of the interaction term (β2) had a *p*-value < 0.05, but the sign of β2 was opposite to that of β1, suggesting that myocardial infarction significantly attenuates the association between pre-ICU MTK exposure and delirium during hospitalization. ^b^ In patients without myocardial infarction, the OR for pre-ICU MTK exposure was 0.586 (i.e., Exp(β1) = 0.586). In patients with myocardial infarction, the OR for pre-ICU MTK exposure was 1.795 (i.e., Exp(β1 + β2) = 1.795).

**Table 3 pharmaceuticals-17-00125-t003:** Association between pre-ICU use of MTK and 90-day mortality mediated by delirium.

Total Effect	*p*-Value	Average Direct Effect (ADE)	*p*-Value	Average Causal Mediation Effect (ACME)	*p*-Value	Percentage of ACME
−0.0645 [−0.1120, −0.0200] ^a^	<0.001	−0.0592 [−0.1069, −0.0100] ^a^	0.020	−0.0053 [−0.0142, 0.0002] ^a^	0.020	8.2%

Adjusted by weight, CCI, APS III, LODS, cerebrovascular disease, paraplegia, severe liver disease, metastatic solid tumor, hemoglobin, hematocrit, platelet, blood urea nitrogen, and sodium. These covariates significantly affected the 90-day mortality and were screened out by the forward stepwise selection method. ^a^ 95% confidence interval.

## Data Availability

The data in this study were extracted from the MIMIC-IV database, which is publicly available and can be found here: https://physionet.org/content/mimiciv/1.0/#files (accessed on 6 April 2023).

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
