# Peer review of "Promising Effects of Montelukast for Critically Ill Asthma Patients via a Reduction in Delirium"

_pharmaceuticals, 2024, doi:10.3390/ph17010125_

Round 1

Reviewer 1 Report

Comments and Suggestions for Authors

This study highlights the potential of MTK and may contribute to the development of novel therapeutic strategies for delirium in critically ill patients. The study is interesting but has some issues, which are as follows:

-Authors need to discuss the possible mechanism for reduction of delirium by MTK. Especially the role of effect of MTK on CysLTR1 in CNS needs to be discussed.

-MTK is usually used in combination with corticosteroids, if it is proposed to be used in other neuropsychiatric patients; will it be used in combination with corticosteroids. Please discuss.

- This reviewer cannot see any figures in the manuscript although they are mentioned in the text. Please provide figures.

Comments on the Quality of English Language

minor

Author Response

Thanks to the comments of the reviewer. 

Our manuscript provides a detailed discussion of the potential mechanisms by which MTK alleviates delirium, with a particular focus on the effects of MTK on CysLTR1 in the CNS. Additionally, we investigated the effects of corticosteroids in patients with psychosis.
The figures and tables have been reattached to the manuscript.

Reviewer 2 Report

Comments and Suggestions for Authors

Manuscript review response: “Promising Effects of Montelukast for Critically Ill Asthma 2 Patients via a Reduction in Delirium”.

Manuscript ID: pharmaceutilcals-2785087

Comments

It is an interesting study on the clinical benefits of montelukas, however it is important to add basic and clinical information to understand the possible mechanisms of action of this medication, through scientific support. You need to add more background and methodology of the study.

Bibliographic citations come before the period. please change.

Introduction

You could add information on in vitro studies of different cell lines, as well as, in animal models, on the effect and action of MTK.

Line 12 add the meaning of the abbreviation of MIMIC-IV.

Line 13 add the meaning of the abbreviation of ICU.

Line 18-20 what is the MTK potential and what is the contribution in the development of novel therapeutic strategies? you need to be more specific in the conclusion.

Line 81 figure 1 is lost.

Line 192 figure 2 is lost.

Line 230 figure 2A is lost.

Line 231 table 2 you could highlight the significance of the p-value.

Line 255 gigure 3 is lost.

Line 265-273 These findings as they compare to other studies.

Line 300 Could you add some perspectives on this? “Require further investigation”.

Line 310-3012 you need add more information about the safety of MTK in 311 pediatric populations.

Conclusion

I suggest a deeper conclusion based on your findings.

Suggesting. I think it is necessary to include more information about asthma and the type of asthma that is being included in this study.

you needs to include a possible (celular or molecular) mechanism of association between asthma andMontelukast maybe a figure or diagram 

Comments on the Quality of English Language

Moderate editing of English language required

Author Response

Thanks to the comments of the reviewer.

We have added the meanings of the acronyms ICU and MIMIIC to the manuscript
Highlighted P-values
Re-added figures and tables
Our study is the first to examine the effect of pre-ICU use of MTK on the incidence of delirium and its prognosis, and to compare it with MTK for other neuropsychiatric disorders.
In our manuscript, we discuss the possible mechanisms by which montelukast reduces delirium and improves prognosis and discuss the safety of MTK in the paediatric population.

Reviewer 3 Report

Comments and Suggestions for Authors

This paper entitled “Promising Effects of Montelukast for Critically Ill Asthma Patients via a Reduction in Delirium” aim to investigate the association between pre-ICU MTK ex- 53 posure and the incidence of delirium, as well as other outcomes, through a retrospective 54 cohort study. The interpretation of the literature contains new ideas of interest to the scientific community. The analysis of the contents and their interpretation are complete and fully adequate to the purposes of the work. The application of the methodologies adopted with respect to the problems addressed is adequate.

Despite this, , I believe that it would benefit from a minor revision

Main concerns:

Abstract: The abstract should start with a brief background and should be divided into chapters: 1) Introduction; 2) Materials and Methods; 3) Results; 4) Discussion; 5) Conclusions.

Introduction: The theoretical framework must be improved. Authors should clearly describe the scientific evidence that supports the hypothesis they have raised.

Conclusions: Authors should add a paragraph where they state the most important outcome of their work

Author Response

Thanks to the reviewers for their comments.

We have re-edited our abstract to highlight our most important findings.

Round 2

Reviewer 1 Report

Comments and Suggestions for Authors

no further comments

Comments on the Quality of English Language

minor

Reviewer 2 Report

Comments and Suggestions for Authors

I don´t have any question.

Comments on the Quality of English Language

 Minor editing of English language required